# Silver nanoparticles from insect wing extract: Biosynthesis and evaluation for antioxidant and antimicrobial potential

**Parameshwar Jakinala[1] , Nageshwar Lingampally[1] , Bee Hameeda [1] *, R. Z. Sayyed[2‡],
Yahya Khan M.[3‡], Elsayed Ahmed Elsayed[4,5‡], Hesham El Enshasy[6,7‡]**

**1** Department of Microbiology, Osmania University, Hyderabad, Telangana, India, **2** Department of
Microbiology, PSGVP Mandal's Arts, Science and Commerce College, Shahada, Maharashtra, India,
**3** Kalam Biotech Pvt. Ltd, Hyderabad, India, **4** Zoology Department, College of Science, King Saud
University, Riyadh, Saudi Arabia, **5** Chemistry of Natural and Microbial Products Department, National
Research Centre, Cairo1, Egypt, **6** Institute of Bioproduct Development (IBD), Universiti Teknologi Malaysia
(UTM), Skudai, Johor Bahru, Malaysia, **7** City of Scientific Research and Technology Applications (CSRTA),
New Burg Al-Arab, Alexandria, Egypt

☯ These authors contributed equally to this work.
‡ These authors also contributed equally to this work.
\* drhami2009@gmail.com

pone.0241729

Institute of Science and Technology, INDIA

**Data Availability Statement:** All relevant data are
within the manuscript file and supplementary file.

## Abstract

Silver nanoparticles (AgNPs) are among the most widely synthesized and used nanoparti-
cles (NPs). AgNPs have been traditionally synthesized from plant extracts, cobwebs, micro-
organisms, etc. However, their synthesis from wing extracts of common insect; *Mang mao*
which is abundantly available in most of the Asian countries has not been explored yet. We
report the synthesis of AgNPs from *M. mao* wings extract and its antioxidant and antimicro-
bial activity. The synthesized AgNPs were spherical, 40–60 nm in size and revealed strong
absorption plasmon band around at 430 nm. Highly crystalline nature of these particles as
determined by Energy-dispersive X-ray analysis and X-ray diffraction further confirmed the
presence of AgNPs. Hydrodynamic size and zeta potential of AgNPs were observed to be
43.9 nm and -7.12 mV, respectively. Fourier-transform infrared spectroscopy analysis
revealed the presence of characteristic amide proteins and aromatic functional groups.
Thin-layer chromatography (TLC) and Gas chromatography-mass spectroscopy (GC-MS)
analysis revealed the presence of fatty acids in the wings extract that may be responsible for
biosynthesis and stabilization of AgNPs. Further, SDS-PAGE of the insect wing extract pro-
tein showed the molecular weight of 49 kDa. *M. mao* silver nanoparticles (MMAgNPs)
exhibit strong antioxidant, broad-range antibacterial and antifungal activities, (66.8 to
87.0%), broad-range antibacterial and antifungal activities was found with maximum zone of
inhibition against *Staphylococcus aureus* MTCC 96 (35±0.4 mm) and *Fusarium oxysporum
f. sp. ricini* (86.6±0.4) which signifies their biomedical and agricultural potential.

## Introduction

Recent past has witnessed a significant dominance of nanotechnology in every field of human
life like biomedical and engineering because it is efficient, bio-friendly, safe and economical

**Funding:** This study was supported by UGC-RGNF in the form of a fellowship (F1-17.1/2014-15/RGNF-2014-15-SC-TEL-86198) to JP, DST-PURSE in the form of funding (C-DST-PURSE-II/43/2018) awarded to HB and the Deanship of Scientific Research at King Saud University through project No. (RG-1440-053) awarded to EAE. Kalam Biotech Pvt. Ltd provided support in the form of a salary for YK. The specific roles of these authors are articulated in the 'author contributions' section. The funders had no role in study design, data collection and analysis, decision to publish, or preparation of the manuscript.

**Competing interests:** The authors have read the journal's policy and have the following competing interests: YK is a Director and owner of Kalam Biotech Pvt. Ltd. This does not alter our adherence to PLOS ONE policies on sharing data and materials. There are no patents, products in development or marketed products associated with this research to declare.

[1]. In recent years, biologically synthesized nanoparticles are preferred over their chemical counterparts [2]. Among various nanoparticles, silver nanoparticles are widely accepted since they can be monitored easily by UV–Vis spectrophotometry [3]. Silver nanoparticles have small size, large surface area, high dispersive ability [4] and exhibit antimicrobial, anticancer, antidiabetic, antioxidant, anti-inflammatory properties [5] and are used in food processing industries, medical implants, ointment fabrication and emulsions [6]. So far, reports on green synthesis of AgNPs are from plant extracts, sea weeds, microorganisms/ metabolites and different biomaterials [7–10] are reported. However, there are no reports from wings of *Mang Mao* insects that are abundantly available and rich in proteins, polysaccharides and lipids. Previous studies show that wings (lipid components) of various insect species show bactericidal mechanisms of nanostructured surfaces [11]. *M. mao* (winged termite) are eusocial insects with three groups (soldiers, workers and queen) and are commonly observed during rainy season (Fig 1A).

After heavy rain, these insects fly out in huge numbers during nights and get assembled near lights, and following day, shed their wings and get killed due to lack of moisture. *Mang Mao* insects are used as edible nutrient-rich and tasty food in most of the Asian countries and in rural areas they are also used to feed chickens, fish, birds and geckos. The wings of dead *M. mao* have a great environmental concern as it is a big waste. Insects' wings can be utilized as an alternative source for chitin and to synthesize nanoparticles [6,8]. Hence, the present study focused on biofabrication of AgNPs by using *M. mao* wings extract and evaluate antioxidant and antimicrobial potential.

## Materials and methods

### Collection and preparation of *Mang mao* wings

*M. mao* wings used in the present study were collected during rainy season from Osmania University (17.4135˚ N, 78.5287˚ E) campus, Hyderabad, India. The wings were collected in sterilized glass beakers, aseptically washed twice with distilled water to remove dust particles, dried and stored at room temperature in air tight containers.

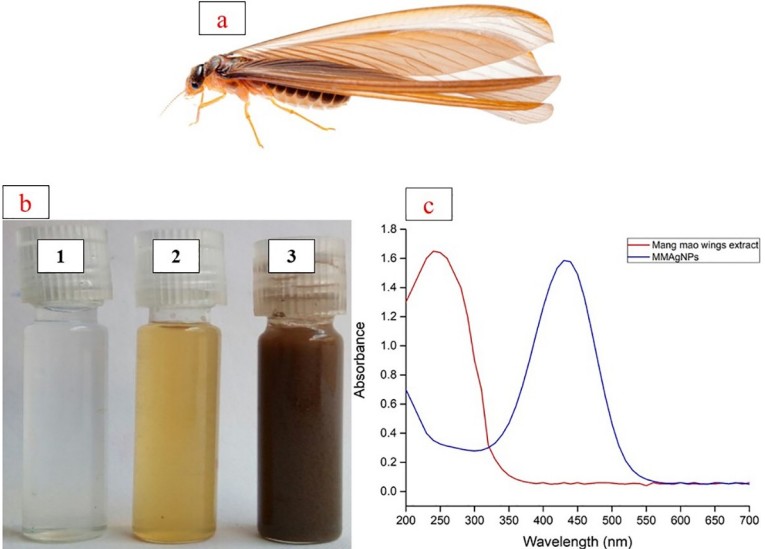

**Fig 1.** (a) *Mang Mao* insect. (b) (1) Silver nitrate solution; (2) *Mang Mao* wings extract; (3) Silver nanoparticles; (c) UV-Vis spectra of AgNPs synthesized by *Mang Mao* wings extract.

## Synthesis of silver nanoparticles (AgNPs)

Biosynthesis of AgNPs from *Mang Mao* wings extract was carried out, following methodology given by Lateef et al. [8] *M. mao* wings (0.1 g) were hydrolyzed using 20 mL of 0.1 M NaOH at 90 $^\circ$C for 1 hour, cooled and the hydrolyzed solution was centrifuged at 8000 rpm for 10 minutes. Supernatant was collected, pH adjusted to 7 from which 1 mL of wing extract was added to 49 mL of 1 mM silver nitrate ($AgNO_3$) solution taken in 100 mL beaker and incubated at 28 ±1˚C for 30 minutes under static conditions for synthesis of AgNPs. Absorption maxima was measured at 200 to 700 nm using UV-Vis spectrophotometer (HITACHI U-2900, Japan) to characterize the *Mang Mao* silver nanoparticles (MMAgNPs) and later on the sample was centrifuged at 8000 rpm for 10 min followed by pellet wash with acetone and air dried for further studies.

## Optimization of MMAgNPs synthesis

Effect of various process parameters on MMAgNPs synthesis was optimized using one variable at a time (OVAT) approach [9] where, the effect of a single parameter was evaluated initially and concentration obtained was used as a standard for all the subsequent steps. Parameters optimized using OVAT included $AgNO_3$ (1 to 3 mM), the concentration ratio of silver nitrate and *M. mao* wings extract (1:1, 1:3, 1:5), pH (3 to 11) and reaction time (0 to 30 minutes). The presence of AgNPs in the resultant solution was detected by the absorbance maxima as mentioned above.

**Stability study.** Above optimized MMAgNPs were incubated at 28±1˚C in dark conditions for 119 days and absorbance maxima was measured weekly once to determine its stability.

## Characterization of MMAgNPs

Scanning electron microscopy (SEM) of the sample was performed by dispersion of the sample in aluminum foil at 5.0 kV operating voltage using JSM-7500F and images were recorded at different magnifications. Energy-dispersive X-ray (EDX) analysis was performed by using Hitachi S-3400 NSEM instrument equipped with Thermo EDX for which the synthesized MMAgNPs were dehydrated and coated on carbon film. X-ray diffraction (XRD) was carried out by drop-coating MMAgNPs solution onto a glass substrate and diffraction was measured using Philips X'Pert Pro X-Ray diffractometer. The average crystal size of nanoparticles was estimated using Scherrer equation i.e., D = K λ/β cosθ [10]. The particle size distribution and zeta potential of MMAgNPs was analyzed in triplicate by electrophoretic light scattering at 25˚C, 150 V (DelsaMax PRO Light Scattering Analyzer, Beckman Coulter, United States) in distilled water. Functional groups were determined by FTIR spectroscopy (Bruker Tensor 27) and spectra were measured in the range 4000–400 cm$^{-1}$ wavelength using KBr pellet as background reference.

Size and zeta potential of the silver nanoparticles were determined by Malvern Zetasizer ZEN 3600 (United Kingdom). This instrument allows the measurement of particle sized distribution in the range 2 nm–3 nm [12].

**Thin layer chromatography and Gas chromatography mass spectroscopy.** Reducing compounds of *M. mao* wings extract was analyzed by TLC on silica gel plates (silica gel 60 F254, Merck, Germany) with mobile phase (chloroform: methanol {97:3}), and chromatogram was examined under UV fluorescence (11). GC-MS analysis of *Mang Mao* wings extract was performed according to method as described by Ha et al. [12].

**Extraction and purification of protein from *M.mao* wings.** The protein from *Mang Mao* wings were extracted as per Zhang et al. [13] and precipitated using ammonium sulfate till approximately 30% saturation and incubated overnight at 4˚C followed by centrifugation. Pellet obtained was suspended in 0.05 M Tris-HCl buffer (pH– 7), with 0.1 M NaCl, and dialyzed

against the same buffer for 24 h. The crude protein was filtered using a 0.22 μm membrane filter and then subjected to column chromatography using silica gel and used for synthesis of AgNPs. Crude and purified proteins were further analyzed by sodium dodecyl sulfate polyacrylamide gel electrophoresis (SDS-PAGE) and its molecular weight (Broad range 11 to 245 kDa, BioLabs, England) was determined [14].

## Antioxidant and antimicrobial activity of MMAgNPs

**DPPH free radical scavenging activity.** Antioxidant activity of MMAgNPs was evaluated by 2, 2-diphenyl-1-picrylhydrazyl (DPPH) free radical scavenging assay [11]. Reaction mixture consisted of equal volumes (1:1 w/v) of different concentrations of the synthesized MMAgNPs (100 to 500 μg mL$^{-1}$ in water), to which 1 mL of 0.5 mM DPPH in ethanol solution was added. The reaction mixture was incubated in dark for 30 min at 28±2˚C. The absorbance is measured at 517 nm by UV-Vis spectrophotometer. Standard used was ascorbic acid to determine scavenging activity which was calculated by following equation.

$$\%\text{scavenging activity} = [(A_{control} - A_{test})/A_{control}] \times 100$$

**Reducing power assay.** Ferric reducing power assay was determined with 1 mL of the synthesized MMAgNPs, 2.5 mL of 0.2 M phosphate buffer (pH–6.6) and 2.5 mL of 1% potassium ferricyanide incubated at 50 $^{o}$C for 20 minutes followed by cooling at room temperature. Then 2.5 mL of 10% trichloroacetic acid was added to the above mix and centrifuged at 8000 rpm for 20 min. Supernatant was mixed with distilled water in 1:1 v/v ratio and 1 mL of 0.1% ferric chloride was added and further incubated at 28± 2˚C for 10 min. Spectrophotometric absorbance of the resultant solution was measured at 700 nm. Increase in absorbance of reaction mixture indicated reducing activity of sample.

**Determination of antibacterial and antifungal activity of MMAgNPS.** Antibacterial activity of MMAgNPs was tested against bacteria namely *Staphylococcus aureus* MTCC 96, *Pseudomonas aeruginosa* MTCC 424, *Escherichia coli* MTCC 43, *Klebsiella pneumonia* MTCC 9751 and *Achromobacter xylosoxidans* SHB 204 (obtained from our lab) and antifungal activity was tested against *Fusarium oxysporum f. sp. ricini*, *Fusarium oxysporum f. sp. lycopersici* MTCC 10270, *Phytophthora nicotianae*, *Fusarium sacchari* and *Colletotrichum falcatum* using agar well diffusion method [15,16] using nutrient and potato dextrose agar plates (Fungal strains were obtained from Indian Institute of Oil Seeds Research (IIOR) Rajendranagar,, Hyderabad, TS, India and Regional Agricultural Research Station (RARS), Anakapalle, AP, India. Aliquots of 50 μL of different concentrations of MMAgNPs (10 μg mL$^{-1}$, 5 μg/mL, 2.5 μg mL$^{-1}$ and 1.25 μg mL$^{-1}$) were separately added in the wells. The inoculated bacterial and fungal plates were incubated at 37˚C for 24 h and 28˚C for 72–96 h respectively and observed for inhibition of growth.

## Statistical analysis

All the experiments were performed in triplicates, repeated twice and data was expressed as means ± standard deviation using IBM Corporation. 2012 Statistics, and graphs were drawn using Origin Pro 2015 [17,18].

## Results and discussion

### Synthesis of MMAgNPs

*M. mao* wings extract, rich in proteins, chitin and lipids was used for the reduction of AgNO$_3$ into AgNPs. The change in color of reaction mixture from yellow to dark brown after 30

minutes incubation (Fig 1) indicated synthesis of MMAgNPs. The intensity of dark brown colour change indicated AgNPs formation from silver salt which is due to excitation of surface plasmon resonance (SPR) effect [19,20].

## Optimization of MMAgNPs synthesis

During the optimization study using OVAT, $AgNO_3$ when used at 1 mM concentration showed maximum absorption at 430 nm, indicated active formation of MMAgNPs. Further increase in the concentration of $AgNO_3$ resulted in decreased absorption (Fig 2A). These results were in agreement with the previous investigations carried out as per Veerasamy et al. [17]. When, *M. mao* wings extract (1 mL) was added to $AgNO_3$ solution (1 mM), rapid conversion to brown color observed within 30 minutes, indicated active MMAgNPs synthesis (Fig 2B). The increased color change of brown color was directly proportional to the incubation period which is due to reduction of $AgNO_3$ and excitation of SPR [19]. The absorption peak varied with different pH and it ranged between 390–470 nm (Fig 2C). The MMAgNPs formation was found to be slow at acidic pH (2–5). At neutral pH (7) absorption maxima was observed at 430 nm and the reaction started as soon as $AgNO_3$ was added to the reaction mixture. The change in color to brown was observed within 30 minutes which indicated MMAgNPs synthesis (Fig 2D).

Further, beyond pH 7 MMAgNPs resulted in aggregation and fall in flocculation. Previous studies revealed that at alkaline pH, Ag (I) ions in solution partly hydrolyze to form

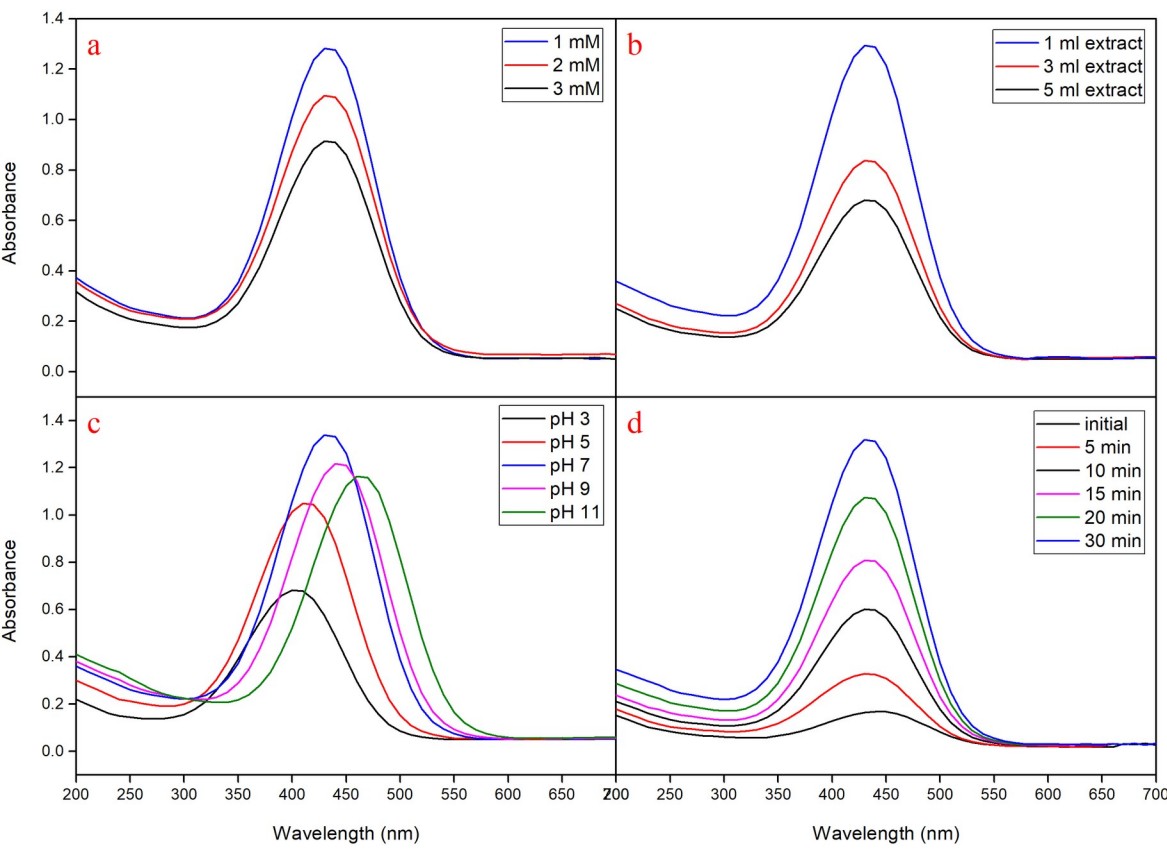

**Fig 2.** (a) UV–vis spectra of aqueous silver nitrate concentration. (b) Concentration ratio of *Mang Mao* extract with 1 mM silver nitrate. (c) Different pH range. (d) Different time intervals.

bioorganic-Ag(OH)x or bioorganic-Ag(NH$_3$)$_2$ complex on the surface of the particle and AgOH/Ag$_2$O colloid in the medium [21]. The characteristic absorption peak at 430 nm corresponded to SPR of AgNPs previously reported by Bahrami-Teimoori et al. [22], which thereby confirmed the synthesis of MMAgNPs. This indicates the significance of *M. mao* wings extract in the reduction of metal salts to their respective metal nano-particles.

**Stability study of MMAgNPs.** The stability of MMAgNPs was monitored up to 119 days (weekly once) and there was no change in absorbance at 430 nm (S1 Fig). This indicated strong stability of biosynthesized MMAgNPs which might be attributed to presence of carboxylate group in proteins that may result in stable nanoparticles [23].

## Characterization of MMAgNPs

**UV–visible spectroscopic analysis of MMAgNPs.** Nanoparticles synthesis is generally found to occur due to excitation of surface plasmon resonance (SPR) [24]. The strongest absorption peak observed at 430 nm (Fig 1C) corresponds to SPR of AgNPs which is in accordance with previous reports [25,26].

**SEM and EDX analysis of MMAgNPs.** SEM analysis revealed the occurrence of MMAgNPs in a spherical shape and their size ranged between 40 to 60 nm. The results showed that variation of pH in reaction mixture altered the nanoparticles size (Fig 3), however, at neutral pH the nanoparticles were uniform (Fig 3C). Reddy et al. [27] reported that biosurfactants (surfactin) used for AgNPs synthesis altered the pH and decreased AgNPs size (9.7 to 4.9 nm) and at pH-9 the nano-particles were uniform. However, MMAgNPs were not uniform in size, and variations in nanoparticle's size were reported by Ahmed et al. [28] and Narayan and Dipak [29] using plant and seaweed extracts. The EDX spectra of MMAgNPs revealed the presence of characteristic signals of silver ions at 3keV (S2 Fig) similar to those observed with chemically synthesized AgNPs. The emission energy at 3 keV indicated the reduction of silver ions to elemental silver [30].

**XRD analysis of MMAgNPs.** The XRD pattern of MMAgNPs showed the diffraction peaks at 2θ values of 32.6˚, 46.57˚, 67.8˚ and 77.04˚ which further confirmed crystalline nature of MMAgNPs (Fig 4) and corresponded to standard JCPDS file No. 04–0783 [31]. Broadening in peaks occurred due to smaller particle size, which reflects the experimental conditions on nucleation and growth of crystal nuclei [32]. According to Debye Scherrer equation, the average nanoparticle size in this study was found to be 32 nm, which was relatively similar as described earlier by Sri Ramkumar et al. [33].

**Zeta potential.** Particle size distribution and zeta potential of MMAgNPs was depicted in S3(A) & S3(B) Fig. Average particle size of synthesized MMAgNPs was 43.9 nm and zeta potential -7.12 mV. The negative charges on AgNPs might be due to *Mang Mao* wings extract covering on nanoparticles. For the determination of overall surface charges on nanoparticles zeta potential analysis was applied.

Stability and prevention of aggregate formation, attributed to repulsions, due to same charge is provided by positive or negative charge on surface of nanoparticles [34]. This indicates better stability of nanoparticles and prevents agglomeration [35]. Hydrodynamic size and zeta potential of MMAgNPs in this study, corroborate with recent report of Badoei-dalfard et al. [36] hydrodynamic size (30–50 nm) of AgNPs synthesized using uricase from *Alcaligenes faecalis* GH3 and zeta potential (-4.6 mV) using *Madhuca longifolia* flower extract [34].

**FTIR analysis.** Identification of bond linkages and functional groups involved in reduction and stability AgNPs synthesis were performed using FTIR. The FTIR spectra of *M. mao* wings extract and synthesis of AgNPs is shown in Fig 5. *M. mao* wings extract bands observed at 3383, 1740, 1641, 1369, 1211, 1049 and 892 cm$^{-1}$, which is related to stretching vibrations of

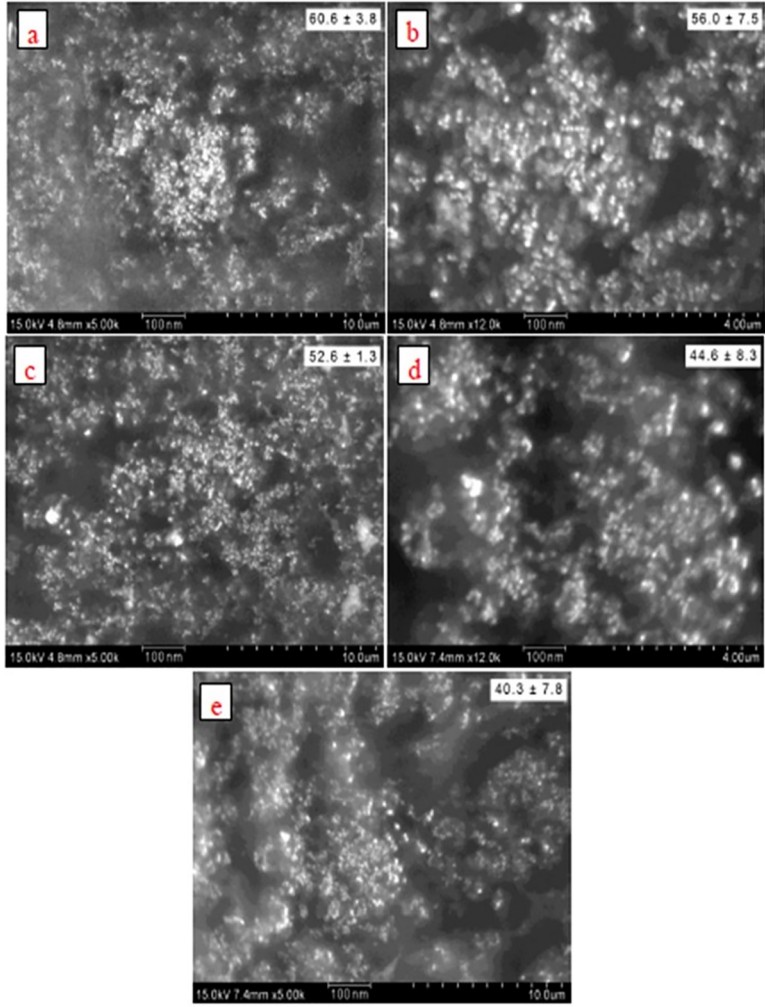

**Fig 3.** SEM images (a) pH-3. (b) pH-5. (c) pH-7. (d) pH-9. (e) pH-11.

OH of carboxylic acids, C = O of ester fatty acid group, C = O of amide band, C-H of aliphatic bending group, C-O-C of polysaccharide, C-O stretching and–C = O of inorganic carbonate respectively.

After reduction, AgNPs form the bands observed at 3291, 2885–2828, 1801, 1643, 1552, 1403, 1322, 1046 and 650 cm$^{-1}$ which is related to stretched vibrations of OH of carboxylic acids, C-H of aliphatics, C = O of anhydride, C = O of amide bond, C = C aromatic, CH of aliphatic bending group, NO$_2$ stretch, C-O stretching and N-H stretch respectively. Biomolecules present in the *M. mao* wings extract might be responsible for synthesis and stability of AgNPs. The characteristic FTIR peaks observed in the present study were similar to those reported by Dhanasekaran et al. [37] and Usha et al. [38]. Similar results were reported by Selvakumar et al. [39] and Soman and Ray [40] where synthesis is performed using *Acalypha hispida* and *Ziziphus oenoplia* (L.) leaf extract as reducing and stabilizing agent.

**TLC of Mang Mao wings extract.**  TLC analysis of *Mang Mao* wings extract corresponded to Rf value 0.96 which is similar to insect chemicals based on previous studies [41]. These chemicals may also be responsible for biofabrication of MMAgNPs (Fig 6A).

**Gas chromatography mass spectrometric analysis.**  GC–MS data revealed that *M.mao* wings extract have 30 major compounds with their molecular weight shown in Table 1. Major

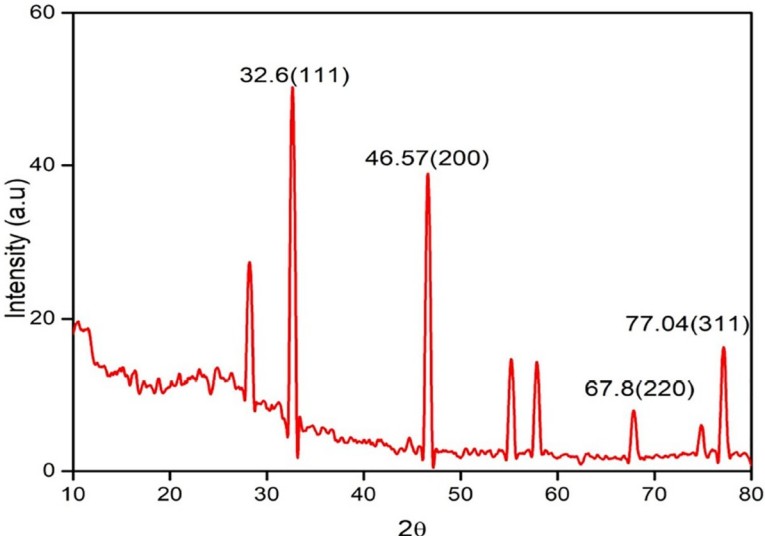

**Fig 4. XRD pattern of biosynthesized silver nanoparticles using *Mang Mao* wings extract.**

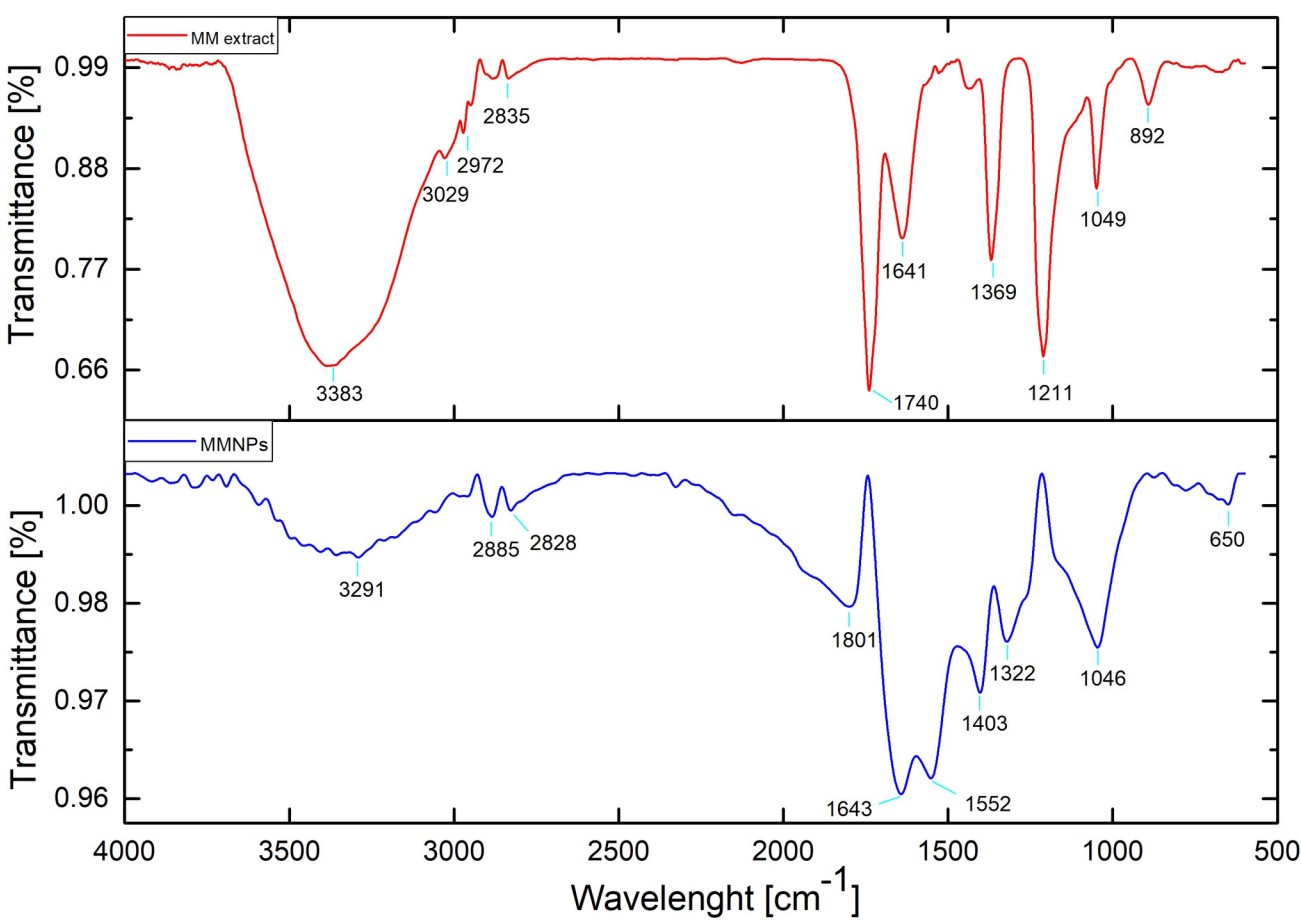

**Fig 5. FTIR pattern of the *Mang Mao* wings extract and synthesized MMAgNPs.**

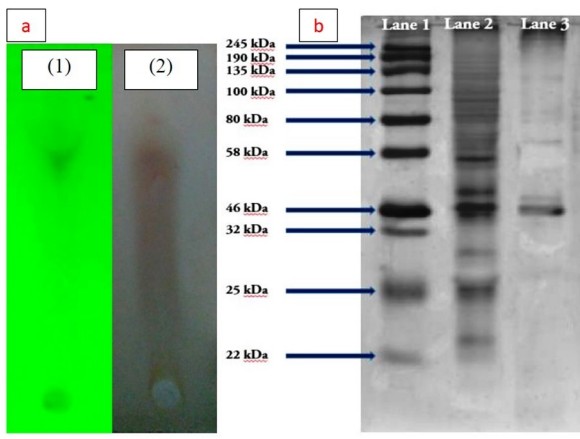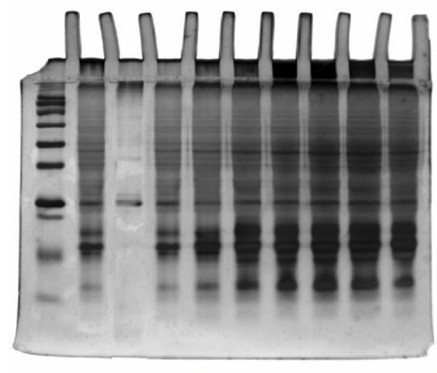

**Fig 6.** (a) TLC of *Mang Mao* wings extract: (1) UV visualization, (2) Ninhydrin reagent. (b) SDS-PAGE analysis of *Mang Mao* wings extract protein; Lane 1. Molecular size marker; lane 2. Crude protein; lane 3. Purified protein (46 kDa) responsible for active biosynthesized MMAgNPs.

components of wings are comprised by aliphatic hydrocarbons and are 9-Octadecynoic acid (16.9%), Nonadecanoic acid (11.4), Ethyl 9-octadecenoate (9.8%), Heptacosanoic acid (4.9%), 1-[2-Deoxy-.beta.-d-erythro-pentofuranosyl] p (4.6%), Hexadecanoic acid (3.7%), 3-Nonyn-2-ol (3.7%), 8-acetoxy-6-benzenesulfonyl-2-th (3.2%), Ethanone (3.1%), 1,5-Cyclooctadiene (3%). These components may be responsible for the reduction and capping of silver ions. Octadecynoic acid and hexadecanoic acid are fatty acids that are widely observed in insects, plants and animals [42–44]. These fatty acids are situated within wings membrane under the epicuticular surface [13,44]. The components present in *Mang Mao* wings extract corresponds with that of previous reports on nanoparticles synthesis [39,45,46].

## Extraction and purification of Mang Mao wings extract

SDS-PAGE analysis revealed the presence of different cellular proteins with molecular weights that ranged between 22–245 kDa. Protein band corresponded to 46 kDa (Fig 6B) was found to act as a capping agent for stabilization of the MMAgNPs.

Khan and Ahmad [47] reported, purified sulfite reductase enzyme with molecular weight of 43 kDa to be responsible for gold nanoparticles stability and synthesis. Kumar et al. [48] also claimed protein of 35.6 kDa is responsible for biosynthesis and capping agent of gold nanoparticles. However, further studies are required towards characterization and identification of this protein to validate this result.

## Antioxidant activity of MMAgNPs

MMAgNPs exhibited DPPH scavenging activity in the range of 66.8 to 87.0% and in case of ascorbic acid it was found to be in the range of 89.7 to 95.5% (Fig 7A). Difference in the activity can be attributed to a different functional group attached to them. MMAgNPs showed good ferric ion reducing activity which was comparable to that of standard ascorbic acid (Fig 7B). Results obtained from this study suggested the use of biosynthesized MMAgNPs as natural antioxidants. Free radical scavenging activity of MMAgNPs is mainly due to the donation of hydrogen molecules, such as proteins, polyphenols and other biomolecules present in the colloidal solution of AgNPs [26]. Radical scavenging activity of AgNPs is due to presence of bioreductant molecules on surface of nanoparticles increasing the surface area for antioxidant activity [12]. Therefore, these MMAgNPs can be employed as a natural antioxidant in the pro-

**Table 1.** GC-MS analysis of *M. mao* wings extract.

| Peak | Retention Time | Peak Area | Area (%) | Peak Height | Base m/z | Compounds present |
|---|---|---|---|---|---|---|
| 1 | 4.293 | 16038 | 2.80 | 5062 | 43.95 | Aspidospermidine-3-carboxylic acid, 2,3-dide |
| 2 | 7.462 | 9307 | 1.63 | 4568 | 43.90 | Histidine, 4-nitro- |
| 3 | 10.765 | 8152 | 1.42 | 4045 | 333.80 | trans-5-Hydroxytricyclo[4.4.0.0(3,8)]-4-carbo |
| 4 | 11.670 | 6678 | 1.17 | 1309 | 42.90 | 5-Pyrimidinecarboxylic acid, hexahydro-5-(1- |
| 5 | 13.051 | 18391 | 3.21 | 4402 | 43.95 | Acetic acid, 8-acetoxy-6-benzenesulfonyl-2-th |
| 6 | 15.405 | 6944 | 1.21 | 3476 | 43.90 | p-Chlorocinnamide |
| 7 | 16.978 | 13566 | 2.37 | 4121 | 58.90 | 2,5,7-Metheno-3H-cyclopenta[a]pentalen-3-o |
| 8 | 17.055 | 17249 | 3.01 | 1863 | 40.00 | 1,5-Cyclooctadiene |
| 9 | 17.287 | 18141 | 3.17 | 5052 | 43.90 | Ethanone, 1-(4-pyridinyl)-, oxime |
| 10 | 17.320 | 26832 | 4.69 | 5182 | 43.90 | 1-[2-Deoxy-.beta.-d-erythro-pentofuranosyl]p |
| 11 | 17.554 | 65501 | 11.44 | 26163 | 88.00 | Nonadecanoic acid, ethyl ester |
| 12 | 19.309 | 96933 | 16.94 | 43045 | 66.95 | 9-Octadecynoic acid |
| 13 | 19.366 | 56171 | 9.81 | 24672 | 54.95 | Ethyl 9-octadecenoate, (E)- |
| 14 | 19.405 | 7072 | 1.24 | 6466 | 43.95 | 3-p-Toluenesulfonyl-7-hydroxymethyl-9-hydr |
| 15 | 19.634 | 21515 | 3.76 | 12272 | 330.11 | Hexadecanoic acid |
| 16 | 21.610 | 12089 | 2.11 | 3085 | 43.95 | Scilliroside |
| 17 | 21.909 | 8632 | 1.51 | 3507 | 66.90 | Cyclohexanol, 2-butyl- |
| 18 | 22.146 | 10102 | 1.77 | 3802 | 43.95 | 6-Benzenesulfonyl-2-oxa-6-aza-adamantane- |
| 19 | 22.980 | 9703 | 1.70 | 6415 | 73.10 | Heptasiloxane, hexadecamethyl- |
| 20 | 23.045 | 10758 | 1.88 | 3399 | 44.00 | N-[2,2,2-Trifluoro-1-(isopropylamino)-1-(trifl |
| 21 | 23.390 | 10906 | 1.91 | 1134 | 43.90 | Carboethoxy-1-piperazinethiocarboxylic acid |
| 22 | 23.520 | 6339 | 1.11 | 2346 | 43.85 | Acetamide, 2,2-dichloro- |
| 23 | 24.200 | 21662 | 3.78 | 3185 | 43.05 | 3-Nonyn-2-ol |
| 24 | 24.830 | 9090 | 1.59 | 4327 | 53.00 | 2-Chloro-3-(chloromethyl)-4-pentenoic acid, |
| 25 | 24.923 | 13120 | 2.29 | 4364 | 41.00 | Butanoic acid, 3-bromo-, ethyl ester |
| 26 | 26.067 | 14088 | 2.46 | 3903 | 42.95 | 3-Cyclopentene-1-propanoic acid, 5-(methox |
| 27 | 26.337 | 28275 | 4.94 | 6648 | 43.05 | Heptacosanoic acid, methyl ester |
| 28 | 26.526 | 7775 | 1.36 | 3530 | 40.10 | 1(2H)-Naphthalenone, 4-ethoxyoctahydro-, tr |
| 29 | 27.795 | 13904 | 2.43 | 3102 | 95.90 | 2-Monooleoylglycerol trimethylsilyl ether |
| 30 | 30.559 | 7400 | 1.29 | 4819 | 73.00 | 3-Isopropoxy-1,1,1,5,5,5-hexamethyl-3-(trime |

oxidants, antioxidants and to balance reactive oxygen species (ROS) levels. Previous study of AgNPs synthesized using *Catharanthus roseus* showed radical scavenging activity and prevented human cell damage and degenerative diseases [49]. This work concludes the use of

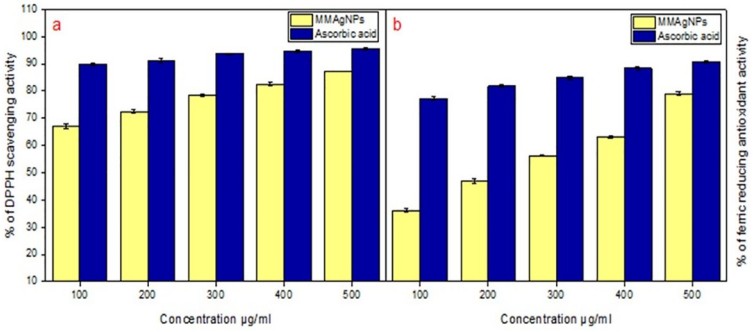

**Fig 7.** Antioxidant activity: (a) DPPH scavenging activity. (b) Ferric reducing antioxidant activity.

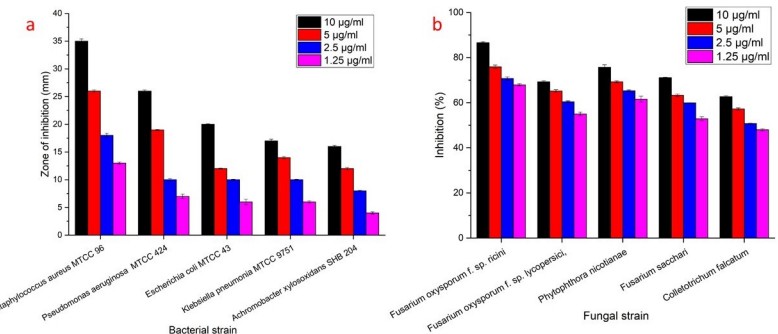

**Fig 8. Antimicrobial activity of synthesized MMAgNPs.**

MMAgNPs as potential agent of antioxidant formulations in biomedical/ pharmaceutical areas.

## Antimicrobial activity of MMAgNPs

Biosynthesized MMAgNPs exhibited potential antibacterial and antifungal activity. MMAgNPs showed maximum zone of inhibition of 35±0.4 mm and minimum zone of inhibition of 16±0.2 mm against *Staphylococcus aureus* MTCC 96 and *Achromobacter xylosoxidans* SHB 204 respectively, when used at a concentration of 10 μg mL$^{-1}$ and the results of the same are shown in Figs 8 and S4. Whereas, maximum percentage of inhibition of 86.6±0.4 mm and minimum percentage of inhibition of 62.7±0.4 mm was recorded against *Fusarium oxysporum f. sp.* ricini and *Colletotrichum falcatum* respectively when the MMAgNPs were used at a concentration of 10 μg mL$^{-1}$. MMAgNPs served as potential antibacterial, antifungal agents and may emerge as an alternative to conventional antibiotics [50].

These AgNPs due to their small size can adhere to bacterial cell membrane, increase permeability and can cause structural changes in bacteria. In case of fungi, AgNPs disrupt the membrane integrity and fungal spores leading to cell death [51]. Some researchers claimed that AgNPs enter the microorganisms and can cause damage by interacting with DNA and proteins, resulting in apoptosis [52].

## Conclusion

In this study, we explored *M. mao* wings extract as an eco-friendly, cost-effective and novel biomaterial for biofabrication of AgNPs. The synthesized AgNPs were spherical, 40–60 nm in size and are highly crystalline nature. This is the first report on use of *M. mao* (seasonal insect) wings with metal chelating potential can be used as natural reducing agents for the synthesis of nanoparticles. MMAgNPs exhibit strong antioxidant (66.8 to 87.0%) and antimicrobial activities was found maximum zone of inhibition against *Staphylococcus aureus* MTCC 96 (35±0.4 mm) and *Fusarium oxysporum f. sp. ricini (*86.6±0.4). Antioxidant and antimicrobial properties of MMAgNPs may further widen their application in the biomedical and agricultural sectors.

## Supporting information

**S1 Fig. Stability of MMAgNPs.**
(DOC)

**S2 Fig. Energy dispersive X-ray analysis of the synthesized MMAgNPs.**
(DOC)

**S3 Fig.** **(a)** Characterization of MMAgNPs by DLS size distribution. **(b)** Characterization of MMAgNPs by zeta potential analysis.
(DOC)

**S4 Fig.** **(a)** Antibacterial activity of MMAgNPs. **(b)** Antifungal activity of MMAgNPs.
(DOC)

**S1 File.**
(PDF)

**S2 File.**
(PDF)

**S1 Raw Image. Gel image for Fig 6.**
(TIF)

# Acknowledgments

Author Parameshwar J and Hameeda Bee are thankful to UGC-RGNF and DST-PURSE. Authors thank Indian Institute of Oil Seeds Research (IIOR) Rajendranagar, TS, India and Regional Agricultural Research Station (RARS), Anakapalle, A.P., India for providing fungal cultures for the present study. The authors extend their appreciation for King Saud University, Riyadh, KSA.

# Author Contributions

**Conceptualization:** Bee Hameeda.

**Formal analysis:** Nageshwar Lingampally, Yahya Khan M.

**Funding acquisition:** Elsayed Ahmed Elsayed.

**Investigation:** Parameshwar Jakinala.

**Methodology:** Parameshwar Jakinala.

**Project administration:** Bee Hameeda.

**Supervision:** Bee Hameeda.

**Writing – review & editing:** R. Z. Sayyed, Elsayed Ahmed Elsayed, Hesham El Enshasy.

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
