## [Decision Letter · Decision Letter 0]

16 Nov 2020

PONE-D-20-32626

Insect wing extract: A novel source for green synthesis of nanoparticles of antioxidant and antimicrobial potential

PLOS ONE

Dear Dr. Bee,

Thank you for submitting your manuscript to PLOS ONE. After careful consideration, we feel that it has merit but does not fully meet PLOS ONE’s publication criteria as it currently stands. Therefore, we invite you to submit a revised version of the manuscript that addresses the points raised during the review process.

We look forward to receiving your revised manuscript.

Kind regards,

Pradeep Kumar

Academic Editor

PLOS ONE

Journal Requirements:

4. Thank you for stating the following in the Financial Disclosure section:

'Elsayed AHmed Elsayed received funds from Deanship of Scientific Research at King Saud University through research group No (RG-1440-053). The funders had no role in study design, data collection and analysis, decision to publish, or preparation of the manuscript.'

We note that one or more of the authors are employed by a commercial company: Kalam Biotech Pvt. Ltd

Reviewers' comments:

Reviewer's Responses to Questions

**Comments to the Author**

1. Is the manuscript technically sound, and do the data support the conclusions?

Reviewer #1: Yes

Reviewer #2: Partly

Reviewer #3: Yes

2. Has the statistical analysis been performed appropriately and rigorously? 

Reviewer #1: I Don't Know

Reviewer #2: I Don't Know

Reviewer #3: N/A

3. Have the authors made all data underlying the findings in their manuscript fully available?

Reviewer #1: Yes

Reviewer #2: Yes

Reviewer #3: Yes

4. Is the manuscript presented in an intelligible fashion and written in standard English?

Reviewer #1: Yes

Reviewer #2: Yes

Reviewer #3: Yes

5. Review Comments to the Author

Reviewer #1: Dear authors,

Great congratulations on publishing this high-quality paper. There are some points should be considered in the final revision of this work.

1-Why the authors select this insect for synthesis of target particles?

2-What is the benefit of applied method in this study?

3-AgNPs are one of the sources of ROS production. How these particles may show such discussed benefits and scavenge free radicals. The provided proofs for this section of experimental assays is not acceptable and the authors must develop their idea regarding the antioxidant activity of AgNPs.

4-For what purpose these particles were synthesized? Would you please kindly discuss the clinical or industrial application of the synthesized particles regarding the method you applied in this study.

5-The authors should determine the synthesized particles are not toxic for human body. Please use MTT assay to determine the potential toxicity effects of these particles for living cells.

6-Figures were not prepared at high-resolution pixels. Please modify the source of figures to provide at least high-quality images for respective readers.

7-Please provide the screenshots of fungal assays as supplementary material. The effectiveness of synthesized particles for inhibition of fungal or bacterial growth on lab plates should be provided.

8-Please discuss the industrial application of the methodology of this paper plus its possible dangers for nearby environment.

Reviewer #2: Dear Author

The manuscript entitled "Insect wing extract: A novel source for green synthesis of nanoparticles of antioxidant and antimicrobial potential" looks interesting and needs to address following queries before considering for publication

Overall Query: Authors needs to justify the importance of insect wings as mentioned by authors "abundantly available and rich in proteins, polysaccharides and lipids" in introduction section. Need to add some imprtant findings and how the nanoparticles synthesized from targeted source will be better than available sources.

Secondaly authors can compare the bioactivities in normal extracts and nanoparticles that will also be interesting.

Minor corrections:

Title : Should be revised as

Silver nanoparticles from Insect wing extract : Biosynthesis and evaluation for antioxidant and antimicrobial potential

2)Abstract

Results of antioxidant and antimicrobial activity need to be mentioned

3)Introduction

Need details various source of biosynthesis of NPs and antioxidant and antimicrobial properties of AgNPs

4)Materials and Methods

Characterization of ------

Brief details of each method should be added

5)Statistical analysis

Cite some reference and source of software

6)Results and Discussion

Figure 4a and 5a and 5b can be given as supplementary figures

7)References

More recent and relevant references can be added

Reviewer #3: This is in reference to the manuscript entitled “Insect wing extract: A novel source for green synthesis of nanoparticles of antioxidant and antimicrobial potential (PONE-D-20-32626) submitted for publication in PLOS ONE journal.

The authors have presented an interesting study concerning green synthesis of nanoparticles. The topic has a novel approach. This is a very valid and interesting study, which can contribute to the body of knowledge. The paper is well-organized. The writing was sufficiently informative to deliver the message of the manuscript. A praiseworthy study was performed by the authors giving sufficient information on producing silver nanoparticles from insect wings.

Hence, I would suggest accepting the paper with minor corrections.

Mention the importance of some significant results obtained in the study in the conclusion section with an emphasis on its future applications.

6. PLOS authors have the option to publish the peer review history of their article (what does this mean?). If published, this will include your full peer review and any attached files.

Reviewer #1: **Yes: **Hassan Rasouli

Reviewer #2: No

Reviewer #3: No

---

## [Author Response · Author response to Decision Letter 0]

1 Jan 2021

Authors are thankful to reviewers for excellent review and comments/suggestions that helped in the improvement of the manuscript. The manuscript is revised in the light of suggestions of reviewers. 

Reviewer #1

Dear authors,

Great congratulations on publishing this high-quality paper. There are some points should be considered in the final revision of this work.

1) Why the authors select this insect for synthesis of target particles?

Authors’ response: There are no reports from wings of Mang Mao insects that are abundantly available. Dead M. mao insects pose disposal and environmental concern and it can be reused for value added product. Recent literature also reveals insect wings has good source for antibacterial activity and now is added in the references.

2) What is the benefit of applied method in this study?

Authors’ response: The methodology used in this study of synthesis of silver nanoparticles, antimicrobial, antioxidant activity of a novel insect source signifies about the biological materials that can be used for nanoparticles and their further applications. In addition it adds on to cost effectiveness, economic feasibility and eco-friendly nature of use of insect wings for nanoparticle synthesis

3) AgNPs are one of the sources of ROS production. How these particles may show such discussed benefits and scavenge free radicals. The provided proofs for this section of experimental assays is not acceptable and the authors must develop their idea regarding the antioxidant activity of AgNPs.

Authors’ response: Agreed. Compounds present in the biological (plant, insect, microbial) extracts play important role in synthesis and stabilization of nanoparticles. These nanoparticles synthesized from insect wing extract showed antioxidant activity due to capped peptides and fatty acids.

4) For what purpose these particles were synthesized? Would you please kindly discuss the clinical or industrial application of the synthesized particles regarding the method you applied in this study.

Authors’ response: The nanoparticles were synthesized to determine their antimicrobial antioxidant potential. The results indicated that these nanoparticles can be used for clinical applications as we these NP exhibited antimicrobial activity against few bacterial pathogens, however, further studies are needed to confirm, their role.

5) The authors should determine the synthesized particles are not toxic for human body. Please use MTT assay to determine the potential toxicity effects of these particles for living cells.

Authors’ response: This study was limited to antimicrobial, cytotoxic and antioxidant aactivities of the synthesized MMAgNP. 

6) Figures were not prepared at high-resolution pixels. Please modify the source of figures to provide at least high-quality images for respective readers.

Authors’ response: Agreed. Resolution of Figures is improved with PLOSONE app and now high resolution figures are included in the paper.

7) Please provide the screenshots of fungal assays as supplementary material. The effectiveness of synthesized particles for inhibition of fungal or bacterial growth on lab plates should be provided.

Authors’ response: Agreed. Screenshots of fungal and bacterial plates are now included as supplementary materials

8) Please discuss the industrial application of the methodology of this paper plus its possible dangers for nearby environment.

Authors’ response: Antioxidant and antimicrobial properties of insect wing extract nanoparticles may further widen their application in the biomedical and agricultural sectors. Use of insect wings for NP synthesis will solve the problems associated with disposal problem and can be better sources for antimicrobial potential, we dont see any negative impact on environment.

Reviewer #2: Dear Author

The manuscript entitled "Insect wing extract: A novel source for green synthesis of nanoparticles of antioxidant and antimicrobial potential" looks interesting and needs to address following queries before considering for publication

1) Authors needs to justify the importance of insect wings as mentioned by authors "abundantly available and rich in proteins, polysaccharides and lipids" in introduction section. Need to add some important findings and how the nanoparticles synthesized from targeted source will be better than available sources.

Authors’ response: The statement in the introduction “Mang mao insect wings are rich in proteins, polysaccharides and lipids” is supported by the observations that revealed the presence of different cellular proteins with molecular weights that ranged between 22-245 kDa. GC MS analysis also revealed the presence of different fatty acids and FTIR analysis reflected the presence of different functional groups present in MMAgNP.

2) Secondly authors can compare the bioactivities in normal extracts and nanoparticles that will also be interesting.

Authors’ response: Agreed. However, we focussed on the evaluation of bioactivities in nanoparticles.

Minor corrections

3) Title : Should be revised as Silver nanoparticles from Insect wing extract : Biosynthesis and evaluation for antioxidant and antimicrobial potential.

Authors’ response: Agreed. Now the title has been revised as Silver nanoparticles from Insect wing extract : Biosynthesis and evaluation for antioxidant and antimicrobial potential 

4) Abstract

Results of antioxidant and antimicrobial activity need to be mentioned

Authors’ response: Results of antioxidant and antimicrobial activity are now included in abstract 

Introduction

5) Need details various source of biosynthesis of NPs and antioxidant and antimicrobial properties of AgNPs

Authors’ response: Now included in MS

6) Materials and Methods 

Characterization of ------

Brief details of each method should be added

Authors’ response: brief details of characterization of nanoparticles of insect wing extract by using various methods is now added in Materials and Methods under characterization of

7) Statistical analysis

Cite some reference and source of software

Authors’ response: Reference number 17 and 18 are now added and source of software i.e. IBM Corp. 2012 Statistics and Origin Pro 2015 and mentioned under subheading Statistical analysis.

Now references are cited and included

6) Results and Discussion

Figure 4a and 5a and 5b can be given as supplementary figures

R: Necessary corrections are made and given as supplementary figures

7) References

More recent and relevant references can be added

R: Now recent and relevant references added in the MS

Reviewer #3: 

This is in reference to the manuscript entitled “Insect wing extract: A novel source for green synthesis of nanoparticles of antioxidant and antimicrobial potential (PONE-D-20-32626) submitted for publication in PLOS ONE journal.

The authors have presented an interesting study concerning green synthesis of nanoparticles. The topic has a novel approach. This is a very valid and interesting study, which can contribute to the body of knowledge. The paper is well-organized. The writing was sufficiently informative to deliver the message of the manuscript. A praiseworthy study was performed by the authors giving sufficient information on producing silver nanoparticles from insect wings.

Hence, I would suggest accepting the paper with minor corrections.

Mention the importance of some significant results obtained in the study in the conclusion section with an emphasis on its future applications.

Authors’ response: Importance of significant results obtained in the study and their possible future applications are now mentioned in the conclusion section.

---

## [Decision Letter · Decision Letter 1]

18 Jan 2021

Silver nanoparticles from Insect wing extract : Biosynthesis and evaluation for antioxidant and antimicrobial potential

PONE-D-20-32626R1

Dear Dr. Bee,

We’re pleased to inform you that your manuscript has been judged scientifically suitable for publication and will be formally accepted for publication once it meets all outstanding technical requirements.

Kind regards,

Pradeep Kumar

Academic Editor

PLOS ONE

Additional Editor Comments (optional):

Reviewers' comments:

Reviewer's Responses to Questions

**Comments to the Author**

1. If the authors have adequately addressed your comments raised in a previous round of review and you feel that this manuscript is now acceptable for publication, you may indicate that here to bypass the “Comments to the Author” section, enter your conflict of interest statement in the “Confidential to Editor” section, and submit your "Accept" recommendation.

Reviewer #1: All comments have been addressed

Reviewer #2: All comments have been addressed

Reviewer #3: All comments have been addressed

2. Is the manuscript technically sound, and do the data support the conclusions?

Reviewer #1: Yes

Reviewer #2: Yes

Reviewer #3: Yes

3. Has the statistical analysis been performed appropriately and rigorously? 

Reviewer #1: Yes

Reviewer #2: Yes

Reviewer #3: Yes

4. Have the authors made all data underlying the findings in their manuscript fully available?

Reviewer #1: Yes

Reviewer #2: Yes

Reviewer #3: Yes

5. Is the manuscript presented in an intelligible fashion and written in standard English?

Reviewer #1: Yes

Reviewer #2: Yes

Reviewer #3: Yes

6. Review Comments to the Author

Reviewer #1: Dear Authors,

Thank you very much for revision of your paper. I've check the content of this paper and it was improved precisely. I have no further comments for evaluation. The manuscript is now ready for publication.

Reviewer #2: Dear Authors

The manuscript is been improved and the authors have addressed all comments given by the reviewers and editors.

Regards

Reviewer #3: The authors have well addressed all the comments. Hence, the article should be considered for publication.

7. PLOS authors have the option to publish the peer review history of their article (what does this mean?). If published, this will include your full peer review and any attached files.

Reviewer #1: **Yes: **Hassan Rasouli

Reviewer #2: No

Reviewer #3: No

---

## [Editor Report · Acceptance letter]

16 Feb 2021

PONE-D-20-32626R1 

Silver nanoparticles from Insect wing extract : Biosynthesis and evaluation for antioxidant and antimicrobial potential 

Dear Dr. Hameeda:

I'm pleased to inform you that your manuscript has been deemed suitable for publication in PLOS ONE. Congratulations! Your manuscript is now with our production department. 

Kind regards, 

on behalf of

Dr. Pradeep Kumar 

Academic Editor

PLOS ONE